# Factors Enhancing Serum Syndecan-1 Concentrations: A Large-Scale Comprehensive Medical Examination

**DOI:** 10.3390/jcm8091320

**Published:** 2019-08-27

**Authors:** Kazumasa Oda, Hideshi Okada, Akio Suzuki, Hiroyuki Tomita, Ryo Kobayashi, Kazuyuki Sumi, Kodai Suzuki, Chihiro Takada, Takuma Ishihara, Keiko Suzuki, Soichiro Kano, Kohei Kondo, Yuki Iwashita, Hirohisa Yano, Ryogen Zaikokuji, So Sampei, Tetsuya Fukuta, Yuichiro Kitagawa, Haruka Okamoto, Takatomo Watanabe, Tomonori Kawaguchi, Takao Kojima, Fumiko Deguchi, Nagisa Miyazaki, Noriaki Yamada, Tomoaki Doi, Takahiro Yoshida, Hiroaki Ushikoshi, Shozo Yoshida, Genzou Takemura, Shinji Ogura

**Affiliations:** 1Department of Emergency and Disaster Medicine, Gifu University Graduate School of Medicine, Gifu 501-1194, Japan; 2Department of Pharmacy, Gifu University Hospital, Gifu 501-1194, Japan; 3Department of Tumor Pathology, Gifu University Graduate School of Medicine, Gifu 501-1194, Japan; 4Innovative and Clinical Research Promotion Center, Gifu University Hospital, Gifu 501-1194, Japan; 5Department of Clinical Laboratory, Gifu University Hospital, Gifu 501-1194, Japan; 6Medical Health Check-up Center, Asahi University Hospital, Gifu 500-8523, Japan; 7Department of Internal Medicine, Asahi University School of Dentistry, Mizuho 501-0223, Japan

**Keywords:** endothelial disorders, glycocalyx injury, syndecan-1, nonlinear regression

## Abstract

Endothelial disorders are related to various diseases. An initial endothelial injury is characterized by endothelial glycocalyx injury. We aimed to evaluate endothelial glycocalyx injury by measuring serum syndecan-1 concentrations in patients during comprehensive medical examinations. A single-center, prospective, observational study was conducted at Asahi University Hospital. The participants enrolled in this study were 1313 patients who underwent comprehensive medical examinations at Asahi University Hospital from January 2018 to June 2018. One patient undergoing hemodialysis was excluded from the study. At enrollment, blood samples were obtained, and study personnel collected demographic and clinical data. No treatments or exposures were conducted except for standard medical examinations and blood sample collection. Laboratory data were obtained by the collection of blood samples at the time of study enrolment. According to nonlinear regression, the concentrations of serum syndecan-1 were significantly related to age (*p* = 0.016), aspartic aminotransferase concentration (AST, *p* = 0.020), blood urea nitrogen concentration (BUN, *p* = 0.013), triglyceride concentration (*p* < 0.001), and hematocrit (*p* = 0.006). These relationships were independent associations. Endothelial glycocalyx injury, which is reflected by serum syndecan-1 concentrations, is related to age, hematocrit, AST concentration, BUN concentration, and triglyceride concentration.

## 1. Introduction

Endothelial disorders are closely related to many diseases, including diabetes mellitus [1], hypertension [2], hypercholesterolemia [3], tumorigenesis [4], ischemia/reperfusion injury [5], respiratory disorder [6], renal dysfunction [7], and autoimmune vasculitis [8]. Previous studies have suggested that treatments used for endothelial disorders could also prevent cardiovascular disease [8,9]. Vascular endothelial disorder exists prior to atherosclerosis, and flow-mediated dilation (FMD) after experimentally imposed increases in shear stress can be used as an index of endothelial function [10,11]. However, FMD is not applicable as a screening approach because it requires the use of echography for diagnosis. Although high-sensitivity C-reactive protein (CRP), lipoprotein-associated phospholipase A2, and pentraxin 3 have been used as biomarkers of endothelial disorder, these markers can only estimate the presence of unstable plaque and do not reflect early vascular endothelial lesions. To date, no biomarkers have been developed to detect early vascular endothelial lesions.

The endothelium lines the inner surface of blood vessels as a thin monolayer and, therefore, is exposed to the circulating blood. Endothelial cells in direct contact with circulating blood are called vascular endothelial cells. Vascular endothelial cells line the entire circulatory system, from the heart to the smallest capillaries. All healthy endothelia are coated by the sugar-protein glycocalyx [12,13,14,15,16,17], which plays key roles in vascular homeostasis, including regulation of microvascular tone and endothelial permeability, maintenance of an oncotic gradient across the endothelial barrier, regulation of the adhesion and migration of leukocytes, and inhibition of intravascular thrombosis [18,19,20]. The glycocalyx is composed of cell-bound proteoglycans, glycosaminoglycan side chains, and sialoproteins [21,22,23]. Proteoglycans consist of a core protein, such as a syndecan family protein, to which glycosaminoglycan is linked. Syndecan-1 is the core protein in heparan sulfate proteoglycan, which is also found in the glycocalyx. Syndecan-1 is released from the endothelium upon injury to the glycocalyx, causing its concentration in the circulation to increase [24]. In fact, serum syndecan-1 was used as an endothelial injury marker in recent clinical studies of sepsis [25,26].

Therefore, in this study, we investigated risk factors for endothelial disorders according to serum syndecan-1 concentrations measured during comprehensive medical examinations.

## 2. Experimental Section

### 2.1. Study Population

In total, 1313 patients who had comprehensive medical examinations at Asahi University Hospital from January 1st, 2018, to June 30th, 2018, participated in this study.

### 2.2. Ethics Approval and Consent to Participate

Ethical permission was obtained from the medical ethics committee of Gifu University Graduate School of Medicine, Gifu, Japan (record no.: 29–214) and Asahi University, Mizuho, Japan (record no.: 30–29), and all patients provided written informed consent.

### 2.3. Consent for Publication

Written informed consent was obtained from the patients for publication of this report.

### 2.4. Clinical Assessments

At enrolment, blood samples were obtained, and study personnel collected demographic and clinical data. Body mass index (BMI) was calculated as weight (kg)/height (m)^2^. Medical and medication histories were obtained from all patients. Patients who received hemodialysis were excluded from the analysis.

### 2.5. Laboratory Data

Laboratory data were obtained by the collection of blood samples at the time of study enrollment more than 12 h after fasting. Serum syndecan-1 concentrations were measured using an enzyme-linked immunosorbent assay (950.640.192, Diaclone, Besancon, Cedex, France).

### 2.6. Statistical Analyses

Descriptive statistics were presented as frequencies and percentages for categorical variables or as medians with interquartile ranges (IQRs) for continuous variables. The primary outcome was serum syndecan-1 concentration. Multivariable regression models were used to assess independent associations between serum syndecan-1 concentrations and blood parameters with adjustment for patient characteristics. Serum syndecan-1 concentrations were natural log-transformed to provide normality in the regression residuals. Beta-coefficients of the regression model were back-transformed to represent the percent increase in serum syndecan-1 concentration with a 50% increase in the corresponding covariate. Nonlinear associations between continuous variables and serum syndecan-1 concentrations were assessed by including nonlinear cubic splines in the regression model. A priori, we chose to include age, sex, BMI, systolic blood pressure, serum total protein, albumin, total bilirubin, aspartic aminotransferase concentration (AST), alanine transaminase, lactate dehydrogenase, blood urea nitrogen (BUN), creatinine concentration, CRP, fasting blood sugar concentration, hemoglobin a1c, serum triglyceride, high-density lipoprotein-cholesterol, low-density lipoprotein-cholesterol, uric acid concentration, hemoglobin concentration, hematocrit (Ht), white blood cell number, and platelet number in the regression model. To avoid bias in the results by excluding missing data, we used multiple imputations in the regression model.

All analyses used a two-sided 5% significance level. Data management and analyses were performed using R version 3.5.1.

## 3. Results

### 3.1. Characteristics of the Patients

Between January and June 2018, we enrolled 1313 patients. One patient undergoing hemodialysis was excluded from the study, thus, we included 1312 patients, with a median age of 51 years (Table 1), in this study. The patients were being treated for hypertension (n = 234, 17.8%), hyperlipidemia (n = 173, 13.2%), diabetes mellitus (n = 80, 6.1%), and hyperuricemia (n = 70, 5.3%). Malignant neoplasms were observed in 65 patients (5.0%; Table 1). Additionally, 85 patients were receiving no treatments and had no abnormal laboratory data.

### 3.2. Associations of Serum Syndecan-1 with Various Parameters

The results of multivariable regression analysis are shown in Table 2. Age, AST concentration, BUN concentration, triglyceride concentration, and Ht were significantly associated with serum syndecan-1 concentrations after adjustment for sex, medication, and BMI. A 1 IQR increase in age was independently associated with a 0.9-fold increase in serum syndecan-1 (*β* = 0.903; 95% confidence interval (CI): 0.831–0.982, *p* = 0.016, Figure 1A). Similar results were found for AST (*β* = 1.093, 95% CI: 0.996–1.200, *p* = 0.020, Figure 1B), BUN (*β* = 1.083, 95% CI: 1.018–1.152, *p* = 0.013, Figure 1C), triglyceride (*β* = 1.131, 95% CI: 1.030–1.242, *p* < 0.001, Figure 1D) and Ht (*β* = 1.726, 95% CI: 1.233–2.417, *p* = 0.006, Figure 1E) after adjustment for covariates. No significant associations were observed for other factors.

According to nonlinear regression analysis, serum syndecan-1 concentrations were significantly related to age, AST concentration, BUN concentration, triglyceride concentration, and Ht. These relationships were independent associations. Significant nonlinear associations were not observed for each variable.

Associations of serum syndecan-1 with (A) Age, (B) AST, (C) BUN, (D) triglyceride, and (E) Ht. AST: aspartate aminotransferase, BUN: blood urea nitrogen, Ht: hematocrit

### 3.3. Subgroup Analysis

Table 3 shows data for the 78 healthy individuals enrolled in this study (that is, individuals receiving no treatment and with no relevant medical history or laboratory data). The median serum syndecan-1 concentration was 19.3 ng/mL (IQR: 13.7–27.3 ng/mL) in healthy participants.

## 4. Discussion

Endothelial disorders are closely related to many diseases via atherosclerosis. The endothelial glycocalyx covers the inner surface of the vascular endothelium and regulates leukocyte adhesion [20], thus, leukocytes cannot adhere to endothelial cells covered with glycocalyx, and endothelial glycocalyx injury may occur prior to atherosclerotic changes. Syndecan-1 is a component of the glycocalyx, and its degradation indicates endothelial injury [24,27,28]. In this study, to detect initial endothelial cell injury, we investigated syndecan-1 concentrations in patients who underwent comprehensive medical examinations. Several previous reports have revealed the relationships of syndecan-1 with severe diseases, such as acute kidney injury, chronic kidney disease, cardiac arrest, cardiovascular disease, and sepsis [29,30,31,32,33]. Although serum syndecan-1 concentrations were reported in several previous studies, the patient populations in these studies were small [29,30,31,32,33,34,35,36]. Additionally, serum syndecan-1 concentrations have not been reported in healthy populations. In this study, 78 healthy individuals with no medication history or abnormal laboratory data were enrolled, and serum syndecan-1 concentrations were determined. However, further studies are required for a more detailed assessment.

The current study revealed that increased serum syndecan-1 concentrations were related to serum triglyceride concentrations. Increased serum triglyceride concentrations may influence vascular endothelial injury and subsequently affect atherosclerosis. Notably, triglycerides increase plasma viscosity [37], affecting fluid shear stress. Since the glycocalyx serves as a mechanosensor for fluid shear stress [38,39,40], fluid shear stress on endothelial cells affects the endothelial glycocalyx [41,42,43], and excess shear stress injures the endothelial glycocalyx. Thus, increasing serum triglyceride concentrations may damage the endothelial glycocalyx directly. Since the current investigation was performed after fasting, serum triglyceride concentrations were not influenced by the consumption of a meal. Similarly, previous reports revealed that elevated Ht increased wall shear stress and affected the degradation of glycocalyx [44,45], and our results also showed that Ht was related to syndecan-1 concentrations.

At the basic research level, there are several studies on the relationship between syndecan-1 and triglyceride [46,47,48,49,50,51]. It was reported that syndecan-1 deficient mice had elevated plasma triglycerides under fasting conditions and mediated hepatic clearance of triglyceride-rich lipoprotein on hepatocytes [46]. Many of these reports showed the function of syndecan-1 on hepatocytes. However, the relationship between serum syndecan-1 and triglyceride concentration is not well-known at the clinical study level. Therefore, to the best of our knowledge, this study is the first to study the relationship between syndecan-1 and triglyceride concentration at the clinical study level.

AST and BUN were also closely related to serum syndecan-1 concentrations in the current study. As syndecan-1 was found to enhance acute kidney injury in a previous report [32], our present findings suggested that the relationship between BUN and syndecan-1 might reflect slight kidney injury. Moreover, increasing BUN is related to dehydration, similar to Ht. AST levels are primarily modulated by liver function. Further studies are needed to elucidate the relationships among AST levels, syndecan-1 concentrations, and liver function.

In this study, we found that age was related to syndecan-1 concentrations. Specifically, with increasing age, syndecan-1 concentrations decreased. This result may be related to the decreased endothelial glycocalyx synthesis of endothelial cells with aging. There are several reports that aging decreases telomerase activity and cell aging with reduction of telomere length. In addition, it is known that aging also increases PAI-1 expression and subsequently decreases vasodilation factor [45,52]. Therefore, there is a possibility that aging affects the decrease of endothelial glycocalyx synthesis.

Our results also showed that increased syndecan-1 concentrations were not related to decreased HbA1c levels. Previous studies have reported that endothelial glycocalyx perturbation is observed in patients with type 1 and type 2 diabetes mellitus [53,54,55,56]. However, in our research, most patients did not have diabetes mellitus, and the experimental settings were different from those of previous studies. Moreover, we evaluated syndecan-1 concentrations as a marker of endothelial glycocalyx injury, whereas previous studies measured glycocalyx volume. Overall, our findings and the results of previous studies suggested that increased syndecan-1 concentrations may reflect the microvessel condition [25,26].

There are many cell adhesion molecules on the cell surface of endothelial cells. Syndecan-1 may potentially be correlated with other biomarkers such as E-selectin and integrin, which are cell adhesion molecules, and might be useful for detecting specific diseases in the future.

This study had some limitations. First, syndecan-1 is expressed not only in the endothelial glycocalyx but also in other organs. However, we did not evaluate the syndecan-1 expression in different organs in this study. Additionally, the endothelial glycocalyx can be injured in response to inflammatory cytokines and other insults. This mechanism was not evaluated in the current study. Further studies are needed to explore these mechanisms. Moreover, the patients included many patients who had been treated for hypertension, hyperlipidemia, diabetes mellitus, and hyperuricemia. However, the detailed treatment information in this medical examination is unknown, while it is known whether the participants received treatment or not.

Although several markers of endothelial disorder have been reported, no biomarkers for extreme initial endothelial injury have been identified to date. Therefore, evaluation of endothelial glycocalyx injury may reveal the initial endothelial injury, and syndecan-1 concentrations may be a biomarker reflecting such damage.

## 5. Conclusions

In conclusion, endothelial glycocalyx injury, which is reflected by serum syndecan-1 concentrations, is related to age, Ht, AST concentration, BUN concentration, and triglyceride concentration. These results suggest that these factors associate with early endothelial injury indicated by endothelial glycocalyx injury. If treatment intervention against these factors is performed as soon as possible, medical expenses can be reduced to quite an extent.

## Figures and Tables

**Figure 1 jcm-08-01320-f001:**
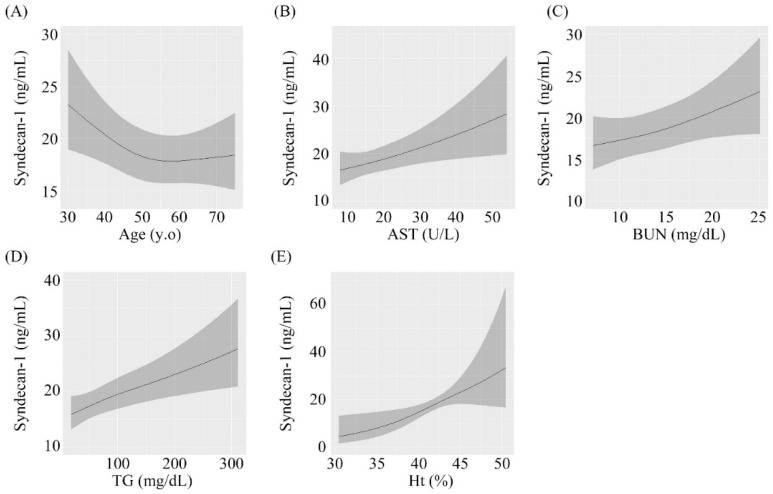
Associations between serum syndecan-1 and different parameters. (**A**) Age, (**B**) Aspartate aminotransferase, (**C**) Blood urea nitrogen, (**D**) Triglyceride, (**E**) Hematocrit.

**Table 1 jcm-08-01320-t001:** Characteristics of the patients.

	Median or Number	(25–75 Percentile)
Number of Cases	1312	
Age	51	(43–59)
Sex (M/F)	819/493	
BMI (kg/m^2^)	22.6	(20.6–24.8)
SBP (mmHg)	122	(111–132)
TP (g/dL)	7.3	(7.1–7.6)
Alb (g/dL)	4.3	(4.2–4.5)
T-Bil (mg/dL)	0.7	(0.5–0.9)
AST (U/L)	17	(14–22)
ALT (U/L)	16	(12–23)
LDH (U/L)	242	(219–268)
BUN (mg/dL)	13.4	(11.4–15.8)
Cre (mg/dL)	0.77	(0.64–0.88)
CRP (mg/dL)	0.04	(0.02–0.09)
FBS (mg/dL)	97	(92–104)
HbA1c (%)	5.4	(5.3–5.6)
TG (mg/dL)	68	(47–99)
HDL –cho (mg/dL)	63	(51–78)
LDL-cho (mg/dL)	115	(97–134)
UA (mg/dL)	5.0	(4.0–6.0)
Hb (g/dL)	14.6	(13.4–15.5)
Ht (%)	42.1	(39.0–44.6)
WBC (×10^3^/µl)	5000	(4200–5900)
Plt (×10^4^/µl)	22.4	(19.3–25.7)
**History of Present Illness**	**Number**	**(Percentage)**
Hypertension	234	(17.8%)
Diabetes Mellitus	80	(6.1%)
Hyperlipidemia	173	(13.2%)
Hyper Uric Acid	70	(5.3%)
Malignant Neoplasm	65	(5.0%)

BMI: Body mass index, SBP: Systolic blood pressure, TP: Total protein, Alb: Albumin, T-Bil: Total bilirubin, AST: Aspartate aminotransferase, ALT: Alanine transaminase, LDH: Lactate dehydrogenase, BUN: Blood urea nitrogen, Cre: Creatinine, CRP: C-reactive protein, FBS: Fasting blood sugar, HbA1c: Hemoglobin A1c, TG: Triglyceride, HDL-Cho: High density lipoprotein-cholesterol, LDL-Cho: Low density lipoprotein-cholesterol, UA: Uric acid, Hb; Hemoglobin, Ht: Hematocrit, WBC: White blood cell, Plt: Platelet.

**Table 2 jcm-08-01320-t002:** Results of multivariable regression analysis.

	25 Percentile	75 Percentile	Fold-Change [IQR]	95%LCI	95%UCI	*p*-Value
Age	43	59	0.903	0.831	0.982	0.016
Sex—Female: Male	-	-	0.883	0.759	1.026	0.105
Medication—Yes: No	-	-	1.030	0.944	1.124	0.505
BMI (kg/m^2^)	20.6	24.8	0.995	0.939	1.054	0.863
SBP (mmHg)	111	132	0.972	0.908	1.039	0.471
TP (g/dL)	7.1	7.6	0.979	0.918	1.044	0.755
Alb (g/dL)	4.2	4.5	0.980	0.913	1.051	0.784
T-Bil (mg/dL)	0.5	0.9	0.940	0.865	1.021	0.168
AST (U/L)	14	22	1.093	0.996	1.200	0.020
ALT (U/L)	12	23	1.046	0.945	1.158	0.484
LDH (U/L)	219	268	0.979	0.916	1.045	0.680
BUN (mg/dL)	11.4	15.8	1.083	1.018	1.152	0.013
Cre (mg/dL)	0.64	0.88	1.048	0.952	1.154	0.591
CRP (mg/dL)	0.02	0.09	1.023	0.935	1.119	0.725
FBS (mg/dL)	92	104	0.993	0.929	1.061	0.603
HbA1c (%)	5.3	5.6	0.954	0.907	1.004	0.077
TG (mg/dL)	47	99	1.131	1.030	1.242	<0.001
HDL-cho (mg/dL)	51	78	1.078	0.989	1.174	0.118
LDL-cho (mg/dL)	97	134	0.970	0.913	1.031	0.515
UA (mg/dL)	4.0	6.0	0.998	0.917	1.087	0.341
Hb (g/dL)	13.4	15.5	0.663	0.470	0.934	0.050
Ht (%)	39.0	44.6	1.726	1.233	2.417	0.006
WBC (×10^3^/µl)	4200	5900	0.976	0.909	1.047	0.666
Plt (×10^4^/µl)	19.3	25.7	1.02	0.960	1.084	0.674

BMI: Body mass index, SBP: Systolic blood pressure, TP: Total protein, Alb: Albumin, T-Bil: Total bilirubin, AST: Aspartate aminotransferase, ALT: Alanine transaminase, LDH: Lactate dehydrogenase, BUN: Blood urea nitrogen, Cre: Creatinine, CRP: C-reactive protein, FBS: Fasting blood sugar, HbA1c: Hemoglobin A1c, TG: Triglyceride, HDL-Cho: High density lipoprotein-cholesterol, LDL-Cho: Low density lipoprotein-cholesterol, UA: Uric Acid, Hb; Hemoglobin, Ht: Hematocrit, WBC: White BLOOD CELL, Plt: Platelet. Fold-changes are derived from the exponential of the *β*-coefficient of the model and represent the fold-change in sensitive serum syndecan-1 accompanying a one interquartile increase in each factor.

**Table 3 jcm-08-01320-t003:** Characteristics of the healthy participants.

	Median or Number	(25–75 Percentile)
Number of Cases	78	
Age	46	(42–52)
Sex (M/F)	41/37	
BMI (kg/m^2^)	21.9	(20.2–22.8)
SBP (mmHg)	110	(102–119)
TP (g/dL)	7.2	(7.0–7.4)
Alb (g/dL)	4.3	(4.1–4.5)
T-Bil (mg/dL)	0.7	(0.5–0.8)
AST (U/L)	16	(13–19)
ALT (U/L)	14	(10–18)
LDH (U/L)	235	(210–255)
BUN (mg/dL)	12.9	(11.0–14.4)
Cre (mg/dL)	0.72	(0.62–0.86)
CRP (mg/dL)	0.03	(0.01–0.05)
FBS (mg/dL)	94	(88–98)
HbA1c (%)	5.3	(5.2–5.5)
TG (mg/dL)	63	(49–84)
HDL-cho (mg/dL)	67	(57–81)
LDL-cho (mg/dL)	105	(90–117)
UA (mg/dL)	4.7	(4.0–5.6)
Hb (g/dL)	14.2	(13.5–15.0)
Ht (%)	41.1	(39.2–43.7)
WBC (×10^3^/µl)	5150	(4525–5875)
Plt (×10^4^/µl)	22.2	(19.0–24.2)

BMI: Body mass index, SBP: Systolic blood pressure, TP: Total protein, Alb: Albumin, T-Bil: Total bilirubin, AST: Aspartate aminotransferase, ALT: Alanine transaminase, LDH: Lactate dehydrogenase, BUN: Blood urea nitrogen, Cre: Creatinine, CRP: C-reactive protein, FBS: Fasting blood sugar, HbA1c: Hemoglobin A1c, TG: Triglyceride, HDL-Cho: High density lipoprotein-cholesterol, LDL-Cho: low density lipoprotein-cholesterol, UA: Uric Acid, Hb; Hemoglobin, Ht: Hematocrit, WBC: White blood cell, Plt: Platelet.

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
