# Peer review of "Factors Enhancing Serum Syndecan-1 Concentrations: A Large-Scale Comprehensive Medical Examination"

_jcm, 2019, doi:10.3390/jcm8091320_

Round 1

Reviewer 1 Report

Overall, the authors present a clear and concise report describing the correlation of soluble E-Selectin with several other indicators of potential endothelial dysfunction.  The authors also provide some perspective of why there is correlation between markers.  The manuscript is well written, the data is presented clearly and appropriate literature is cited.  A few minor revisions would improve the impact of the paper in my opinion.

1) While the average selectin concentration is given for healthy patients, I could not find this number in the report for other patients.  Is it available?

2) I fully agree that early detection of glycocalyx damage and endothelial dysfunction is important.  Is these any possibility of speculating how detection of soluble e-selectin and potentially its correlation with other biomarkers could lead to specific disease detection, or appropriate treatment for patients?

3) The first few lines of the results section should be removed as these are general guidelines form the publisher.

Author Response

Response to Reviewer #1:

Reviewer #1’s useful and insightful comments are appreciated. Responses to the comments are below.

1) While the average selectin concentration is given for healthy patients, I could not find this number in the report for other patients.  Is it available?

2) I fully agree that early detection of glycocalyx damage and endothelial dysfunction is important.  Is these any possibility of speculating how detection of soluble e-selectin and potentially its correlation with other biomarkers could lead to specific disease detection, or appropriate treatment for patients?

--------------------------------------------------------------------------------------------------------------------

We did not provide any data for selectin in our submitted manuscript. On the other hand, we agree with Reviewer #1 that assessment of selectin concentration is critical for this manuscript. We have added new sentences about selectin in the discussion section of the revised manuscript as follows.

Discussion Line 210-212

There are many cell adhesion molecules on the cell surface of endothelial cells. Syndecan-1 may potentially be correlated with other biomarkers such as E-selectin and integrin, which are cell adhesion molecules, and might be useful for detecting specific diseases in the future.

3) The first few lines of the results section should be removed as these are general guidelines form the publisher.

--------------------------------------------------------------------------------------------------------------------

We have deleted the indicated text.

Reviewer 2 Report

In this study, the authors aim at using syndecan-1 as a marker to detect early endothelial injury. Serum syndecan-1 concentrations and other blood parameters were obtained from 1312 patients. Syndecan-1 concentration was found to associate with age, AST, BUN triglyceride and hematocrit. The authors concluded that these factors cause early endothelial injury.

Hypertension, diabetes, and hyperlipidemia are known risk factors for endothelial injury. What was the syndecan-1 concentration in these subgroups? Were they significant different from the healthy enrollments?

Have any patients included in this study ever been diagnosed to have atherosclerosis?

Vascular shear stress is influenced by various factors such as viscosity, hematocrit and hemodynamic factors. Since the hemodynamic data was not observed in this study, it is difficult to determine the effect of triglyceride concentration on shear stress. However, since hyperlipidemia is a major risk factor for endothelial injury, is there other evidence or literature on the relationship on syndecan-1 and triglyceride?

Discussion Lines 187-188, the sentence is confusing. Did the authors suggest that elevated syndecan-1 may cause kidney injury or otherwise?

The authors suggest that endothelial glycocalyx synthesis is decreased with aging. Is there any literature that can support this claim?

Author Response

Response to Reviewer #2:

We are grateful for Reviewer #2’s useful and insightful comments. Responses to the reviewer’s comments are detailed below.

Hypertension, diabetes, and hyperlipidemia are known risk factors for endothelial injury. What was the syndecan-1 concentration in these subgroups? Were they significant different from the healthy enrollments?

--------------------------------------------------------------------------------------------------------------------

Based on the reviewer’s suggestion, we have prepared a Reviewer Table as shown below.

In these results, serum syndecan concentration was not significantly different among healthy participants, hypertension, diabetes, hyperlipidemia, and hyperuricemia patients. One of the reasons is that these patients included many patients who had been treated for hypertension, hyperlipidemia, diabetes mellitus, and hyperuricemia. To perform a more accurate analysis, detailed information on the treatments is required. However, the detailed treatment information in this medical examination is unknown, while it is known whether the participants received treatment or not. Since we are afraid that the insufficient information may be misleading for readers, we have chosen to show this data to the Reviewers only. In addition, we added this information in the revised manuscript as a limitation.

Discussion Line 217-220

Moreover, the patients included many patients who had been treated for hypertension, hyperlipidemia, diabetes mellitus, and hyperuricemia. However, the detailed treatment information in this medical examination is unknown, while it is known whether the participants received treatment or not.

Reviewer Table: Summary of Syndecan-1 by the subgroup of interest

n

Mean

Median

25 percentile

75 percentile

p-value

Healthy

78

24.9

19.3

13.7

27.3

-

Hypertension

234

27.3

19.9

13.6

30.4

0.915

Diabetes

80

23.5

18.3

12.3

28.3

0.823

Hyperlipidemia

173

24.2

18.7

12.5

26.3

0.991

Hyperuricemia

70

38.2

26.2

15.9

43.3

0.576

P-values were obtained by Wilcoxon rank sum test comparing healthy participants and each subgroup.

Have any patients included in this study ever been diagnosed to have atherosclerosis?

--------------------------------------------------------------------------------------------------------------------

We appreciate the Reviewer’s comment.

We certainly thought that some atherosclerosis patients were included in this study. However, methods for diagnosis of atherosclerosis such as carotid artery ultrasonography were not performed in this medical checkup. Therefore, we cannot provide exact numbers.

The medical checkup forms only one of the screening examinations and the examination items are not that many.

Vascular shear stress is influenced by various factors such as viscosity, hematocrit and hemodynamic factors. Since the hemodynamic data was not observed in this study, it is difficult to determine the effect of triglyceride concentration on shear stress. However, since hyperlipidemia is a major risk factor for endothelial injury, is there other evidence or literature on the relationship on syndecan-1 and triglyceride?

--------------------------------------------------------------------------------------------------------------------

At the basic research level, there are several studies on the relationship between syndecan-1 and triglyceride. It was reported that syndecan-1 deficient mice had elevated plasma triglycerides under fasting conditions and mediated hepatic clearance of triglyceride-rich lipoprotein on hepatocytes. Many of these reports showed the function of syndecan-1 on hepatocytes. However, the relationship between serum syndecan-1 and triglyceride concentration is not well-known at the clinical study level.

We have added new sentences and references to the revised manuscript as follows.

Discussion Lines 183-189

At the basic research level, there are several studies on the relationship between syndecan-1 and triglyceride [46-51]. It was reported that syndecan-1 deficient mice had elevated plasma triglycerides under fasting conditions and mediated hepatic clearance of triglyceride-rich lipoprotein on hepatocytes [46]. Many of these reports showed the function of syndecan-1 on hepatocytes. However, the relationship between serum syndecan-1 and triglyceride concentration is not well-known at the clinical study level. Therefore, to the best of our knowledge, this study is the first to study the relationship between syndecan-1 and triglyceride concentration at the clinical study level

Reference Section

Stanford, K.I.; Bishop, J.R.; Foley, E.M.; Gonzales, J.C.; Niesman, I.R.; Witztum, J.L.; Esko, J. D. Syndecan-1 is the primary heparan sulfate proteoglycan mediating hepatic clearance of triglyceride-rich lipoproteins in mice. J. Clin. Invest. 2009, 119, 3236-3245. Fuki, I. V.; Kuhn, K.M.; Lomazov, I.R.; Rothman, V.L.; Tuszynski, G.P.; Iozzo, R.V.; Swenson, T.L.; Fisher, E.A.; Williams. K. J. The syndecan family of proteoglycans. Novel receptors mediating internalization of atherogenic lipoproteins in vitro. J. Clin. Invest. 1997, 100, 1611-1622. Zeng, B.J.; Mortimer, B.C.; Martins, I.J.; Seydel, U.; Redgrave, T.G. Chylomicron remnant uptake is regulated by the expression and function of heparan sulfate proteoglycan in hepatocytes. J. Lipid Res. 1998 39, 845-860. Yu, K.C.; Chen, W.; Cooper, A.D. LDL receptor-related protein mediates cell-surface clustering and hepatic sequestration of chylomicron remnants in LDLR-deficient mice. J. Clin. Invest. 2001, 107, 1387-1394. Meyer, M.E.; Williams, K.J. Transmembrane and cytoplasmic domains of syndecan mediate a multi-step endocytic pathway involving detergent-insoluble membrane rafts. Biochem. J. 2000, 351, 607–612. Cortés, V.; Amigo, L.; Donoso, K.; Valencia, I.; Quiñones, V.; Zanlungo, S.; Brandan, E.; Rigotti, A. Adenovirus-mediated hepatic syndecan-1 overexpression induces hepatocyte proliferation and hyperlipidaemia in mice. Liver Int. 2007, 27, 569–581.

Discussion Lines 187-188, the sentence is confusing. Did the authors suggest that elevated syndecan-1 may cause kidney injury or otherwise?

--------------------------------------------------------------------------------------------------------------------

We apologize for the confusion. We have revised the sentences as follows.

Discussion Line 191-193

Because syndecan-1 was found to enhance acute kidney injury in a previous report [32], our present findings suggested that the relationship between BUN and syndecan-1 might reflect slight kidney injury.

The authors suggest that endothelial glycocalyx synthesis is decreased with aging. Is there any literature that can support this claim?

--------------------------------------------------------------------------------------------------------------------

There are several reports that aging decreases telomerase activity and cell aging with reduction of telomere length. In addition, it is known that aging also increases PAI-1 expression and subsequently decreases vasodilation factor.

Therefore, although, it is just a speculation, there is a possibility that aging affects the decrease of endothelial glycocalyx synthesis. We have added new sentences and references in the revised manuscript as follows.

Discussion Line 198-201

There are several reports that aging decreases telomerase activity and cell aging with reduction of telomere length. In addition, it is known that aging also increases PAI-1 expression and subsequently decreases vasodilation factor [45 52]. Therefore, there is a possibility that aging affects the decrease of endothelial glycocalyx synthesis

Reference Section

Chang, E.; Harley, C. B. Telomere length and replicative aging in human vascular tissues. Proc. Natl. Acad. Sci. U.S.A. 1995, 92, 11190–11194.

52. Sherr, C.J.; DePinho, R. A. Cellular senescence: mitotic clock or culture shock? Cell 2000, 102, 407-410.

Round 2

Reviewer 2 Report

According to the information provided by the authors, it remains unclear the relationships between syndecan-1 concentration and AST, BUN and TG. Thus, the statement in the Conclusion section: “…these factors cause early endothelial injury indicated by endothelial glycocalyx injury” may be far-fetched. I would suggest using “associate” instead of “cause” in the conclusion.

Author Response

Response to Reviewer #2:

We are grateful for Reviewer #2’s useful comments. Responses to the reviewer’s comments are detailed below.

According to the information provided by the authors, it remains unclear the relationships between syndecan-1 concentration and AST, BUN and TG. Thus, the statement in the Conclusion section: “…these factors cause early endothelial injury indicated by endothelial glycocalyx injury” may be far-fetched. I would suggest using “associate” instead of “cause” in the conclusion.

--------------------------------------------------------------------------------------------------------------------

We have revised the sentences as follows.

Conclusion Line 228-229

These results suggest that these factors associate with early endothelial injury indicated by endothelial glycocalyx injury.
